# Contained *Mycobacterium tuberculosis* infection induces concomitant and heterologous protection

**Johannes Nemeth**[1¤], **Gregory S. Olson**[1,2], **Alissa C. Rothchild**[1], **Ana N. Jahn**[1], **Dat Mai**[1], **Fergal J. Duffy**[1], **Jared L. Delahaye**[1], **Sanjay Srivatsan**[2,3], **Courtney R. Plumlee**[1], **Kevin B. Urdahl**[1], **Elizabeth S. Gold**[1], **Alan Aderem**[1☯], **Alan H. Diercks**[1☯]*

1 Seattle Children's Research Institute, Seattle, Washington, United States of America, 2 Medical Scientist Training Program, University of Washington School of Medicine, Seattle, Washington, United States of America, 3 Department of Genome Sciences, University of Washington, Seattle, Washington, United States of America

☯ These authors contributed equally to this work.
¤ Current address: Division of Infectious Diseases and Hospital Epidemiology, University Hospital Zurich, Zurich, Switzerland
* alan.diercks@seattlechildrens.org

**Data Availability Statement:** Raw and processed transcriptomic data are deposited in GEO (GSE126355).

## Abstract

Progress in tuberculosis vaccine development is hampered by an incomplete understanding of the immune mechanisms that protect against infection with *Mycobacterium tuberculosis* (Mtb), the causative agent of tuberculosis. Although the M72/ASOE1 trial yielded encouraging results (54% efficacy in subjects with prior exposure to Mtb), a highly effective vaccine against adult tuberculosis remains elusive. We show that in a mouse model, establishment of a contained and persistent yet non-pathogenic infection with Mtb ("contained Mtb infection", CMTB) rapidly and durably reduces tuberculosis disease burden after re-exposure through aerosol challenge. Protection is associated with elevated activation of alveolar macrophages, the first cells that respond to inhaled Mtb, and accelerated recruitment of Mtb-specific T cells to the lung parenchyma. Systems approaches, as well as *ex vivo* functional assays and *in vivo* infection experiments, demonstrate that CMTB reconfigures tissue resident alveolar macrophages via low grade interferon-γ exposure. These studies demonstrate that under certain circumstances, the continuous interaction of the immune system with Mtb is beneficial to the host by maintaining elevated innate immune responses.

## Author summary

Paradoxically, although tuberculosis (TB) ranks as the deadliest infectious disease worldwide, the immune mechanisms that protect against the disease are quite effective: Despite a high prevalence of infection with *Mycobacterium tuberculosis* (Mtb), the vast majority of individuals with an intact immune system contain the infection indefinitely with no clinical symptoms. Historical cohort studies and contemporary epidemiological studies indicate that prior infection with Mtb is actually protective against the development of active

**Funding:** Research reported in this publication was supported by the National Institute of Allergy and Infectious Diseases (Grant numbers U19AI135976, U19AI100627, and R01AI032972, to AA, www.nih. gov). JN was supported by Swiss National Science Foundation (SNSF, http://www.snf.ch/en/Pages/ default.aspx) grant #P300PB_164742. The funders had no role in study design, data collection and analysis, decision to publish, or preparation of the manuscript.

**Competing interests:** The authors have declared that no competing interests exist.

TB after re-exposure. Understanding the mechanisms underlying this natural protection would inform vaccine design efforts, however progress has been hampered by the lack of a small animal model of the protective effects of contained Mtb infection (CMTB). Previously, the protective effects have been attributed to adaptive immune responses. This study shows that CMTB also affects the innate immune response and is associated with low-level interferon-γ cytokinemia. While experiments in mice have elucidated many of the fundamental mechanisms underlying the immune response to Mtb, a small-animal model for the protective effect of CMTB, a critical feature of the human disease, has been elusive. Here, we demonstrate a mouse model that can enable mechanistic studies of the well-established but poorly understood role of CMTB in protection against re-infection.

## Introduction

Tuberculosis (TB) ranks as the deadliest infectious disease world-wide with an estimated 10.4 million cases of active disease and 1.8 million deaths annually [1]. The failure to identify mechanisms or correlates of protection against TB has hampered the development of vaccines or host-directed therapies to reduce the world-wide disease burden

It is widely believed that the clinical definition of TB encompasses a spectrum of disease states including bacterial clearance, persistence of live bacteria contained by the immune system, and sub-clinical and overt disease [2]. It is clear that humans may harbor viable Mtb in a variety of tissues for decades without developing symptoms [3–6]. In both humans [7] and non-human primates [8], the lymphatic system plays a key role by providing a reservoir for latent Mtb [3,9] and coordinating subsequent immune responses.

The immune mechanisms that protect against TB are evidently quite powerful: despite an extraordinarily high prevalence of *Mycobacterium tuberculosis* (Mtb) (some estimates suggest that at least 25% of the world's population has been exposed [10]), the vast majority (~90%) of individuals with an intact immune system are able to contain and control the infection for their lifetimes with no clinical symptoms [10–12].

Both historical cohort studies and contemporary epidemiological studies demonstrate that prior infection is protective against re-infection [13,14]. In a non-human primate model, prior infection with Mtb conferred significant protection against aerosol challenge [15]. The phenomenon of a low-grade infection protecting against subsequent infections with the same pathogen has led to the development of almost all live vaccines currently in use, including those based on bacteria (BCG) or experimental vaccines against parasites (Leishmania) [16]. Likewise, continuous exposure to *Plasmodium falciparum*, the causative agent of malaria, and maintenance of low-grade parasitemia provides protection against high-density parasitemia and death in adults [17]. This phenomenon is often referred to as "concomitant immunity" [18] or, in the case of malaria, "premunition" [17]. Despite the strong evidence that prior infection with Mtb confers protection against reinfection, the underlying mechanisms have not been defined, in part due to the lack of a suitable small animal model.

Several recent studies have shown that vaccination with live bacteria (BCG) contributes to protection against TB disease, in part, by altering the activation state of the innate immune system. For example, protection against aerosol challenge with Mtb conferred by intravenous BCG vaccination was linked to epigenetic reprogramming of bone marrow derived macrophages in mice [19]. Additionally, BCG vaccination was able to prevent immune pathology and bacterial dissemination independently of CD4 T cells in a mouse model of TB reactivation [20].

Recent work in mice has demonstrated that virulent Mtb which are injected into the dermis of the ear are asymptomatically contained in the draining lymph node for an extended period of time (Contained MTB infection model, CMTB). This model was used to mimic failure of containment arising from immune suppression and predicted a novel cell type as a niche for Mtb persistence in humans [21,22].

Because the CMTB mouse model shares many features of asymptomatic Mtb infection in humans, we hypothesized that it reflects a portion of the Mtb-infection spectrum and would provide protection against active TB disease. Here, we demonstrate that CMTB in mice confers strong protection against both aerosol challenge with Mtb and heterologous challenges and is associated with amplified innate immune activation that is dependent on low-levels of circulating interferon-γ.

## Results

### Intradermal Mtb-infection of mice recapitulates key aspects of human CMTB and is stable for up to 1 year

To establish a system in which to examine the protective effect of CMTB, we adopted a model that was originally developed to study mechanisms of Mtb containment [21]. We infected mice intradermally in the ear with 10,000 CFU of a commonly used virulent strain of Mtb, H37Rv [21]. Within 5 days, the bacteria trafficked to the ipsilateral superficial cervical lymph nodes which, when extracted, were visibly enlarged. The bacterial burden in the draining lymph node was relatively stable for at least one year (Fig 1A) with minimal dissemination to the spleen (S1 Fig) and no detectable dissemination to the lung (detection limit 10 CFU/lung). We measured the circulating levels of 38 cytokines, and detected low amounts of circulating CXCL10 (mean 67 pg/mL, range 43–98 pg/mL), interferon-γ (IFNG) (mean 5.3 pg/mL, range 0.6–11.2 pg/mL), and IL6 (mean 7.2 pg/mL, range 0–14.4 pg/mL) in the first 6 weeks following the establishment of CMTB (Fig 1B). The serum concentrations of the remaining analytes measured were not substantially affected by CMTB (S2 Fig). These serum levels of IFNG are 10 to 100 times lower than those in mouse models of active TB disease [23,24]. This cytokine pattern mirrors reported measurements in humans; asymptomatically Mtb-infected individuals typically exhibit circulating IFNG levels in the low pg range [25] while those with active disease have cytokinemia 10–100 times higher [26].

In humans, asymptomatic Mtb-infection is clinically defined by the presence of an Mtb-specific T cell response in the absence of active disease. In CMTB mice, Mtb-specific CD8 T cells were detectable in the circulation from day 10 onward without recruitment to the lung parenchyma (Fig 1C). However, we were unable to detect ESAT-6 specific CD4 T cells in peripheral blood, lung or spleen. Furthermore, for at least one year, the mice displayed no overt systemic symptoms (e.g. weight loss or changes in coat or behavior) nor any local symptoms (including visible inflammation or irritation). Taken together, these data indicate that intradermal Mtb-infection of mice recapitulates many key aspects of asymptomatic Mtb-infection in humans, including circulating Mtb-specific T cells in the absence of disease symptoms.

### CMTB mice are strongly protected against aerosol infection with Mtb and heterologous challenges

To test the hypothesis that CMTB is protective against aerosol infection, we challenged mice with 50–100 CFU of Mtb H37Rv 8–10 weeks after the establishment of CMTB and measured bacterial burden in the lung and spleen at 14, 42, and 100 days after infection. At each of these time points, the bacterial burden in both tissues was significantly lower in CMTB mice

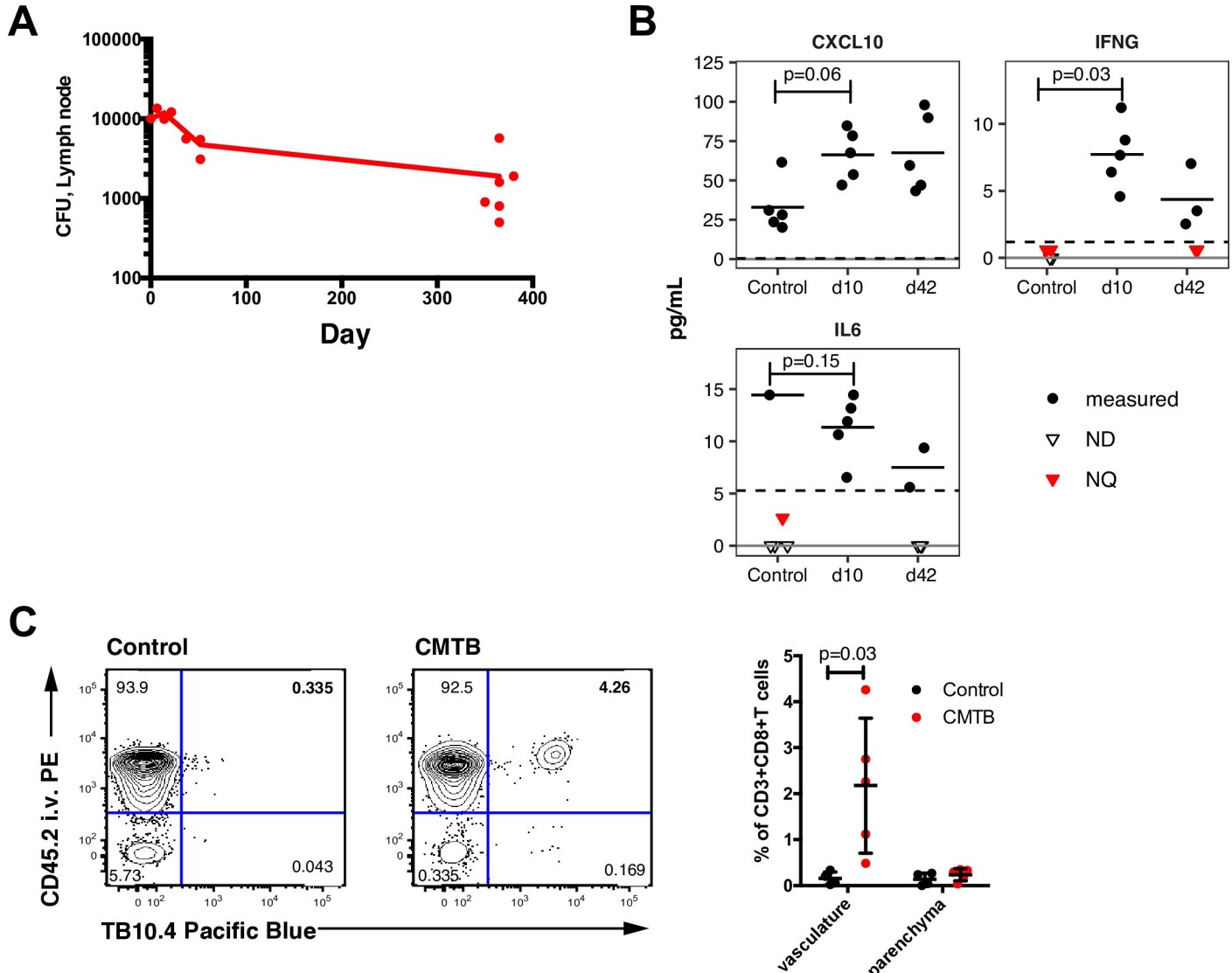

**Fig 1. CMTB mice have an altered inflammatory milieu.** (**A**) Mice were inoculated intradermally in the ear with 10,000 CFU of H37Rv Mtb and bacterial burden in the superficial cervical lymph nodes measured 10, 42, and 365 days following infection by CFU assay. Data are a representative experiment of 2 independent experiments with 3–6 mice/timepoint. (**B**) CMTB was established as described in (**A**) and the levels of the indicated cytokines in peripheral blood were measured by multiplexed immunoassay (Luminex) at the indicated time points. ND = below detection limit; NQ = below quantification limit. Dashed line indicates the quantification limit. Out of 38 cytokines/chemokines assayed, only CXCL10, IFNG, and IL6 exhibited CMTB-induced concentration changes (at B-H corrected p < 0.2). For statistical calculations, values below the detection limit were set to zero and values below the quantification limit were set to half of the quantification limit. (**C**) Relative proportions of CD3+CD8+TB10.4-tetramer-positive cells in the lung vasculature (CD45.2+) and parenchyma (CD45.2-) determined 42 days after the establishment of CMTB by flow cytometry. Localization to the vasculature or parenchyma was determined by i.v. labeling with an anti-CD45.2 antibody 10 minutes prior to blood collection. Data are representative of 2 experiments with *n* = 4–5 mice per experiment. Statistical significance was determined by Student's *t*-test. Error bars depict the mean and SD. (See S16 Fig for gating strategy).

compared to controls (Fig 2A). We obtained similar results when mice were challenged 10 to 14 days after establishment of CMTB (S3 Fig): CMTB mice had on average 18.4-fold (CI: 10.6–26.3) fewer bacteria in the lung as compared to controls measured 6 weeks after aerosol challenge across 6 independent experiments with a total of 60 mice. This level of protection exceeds that observed in the standard model of BCG vaccination in mice (sub-cutaneous

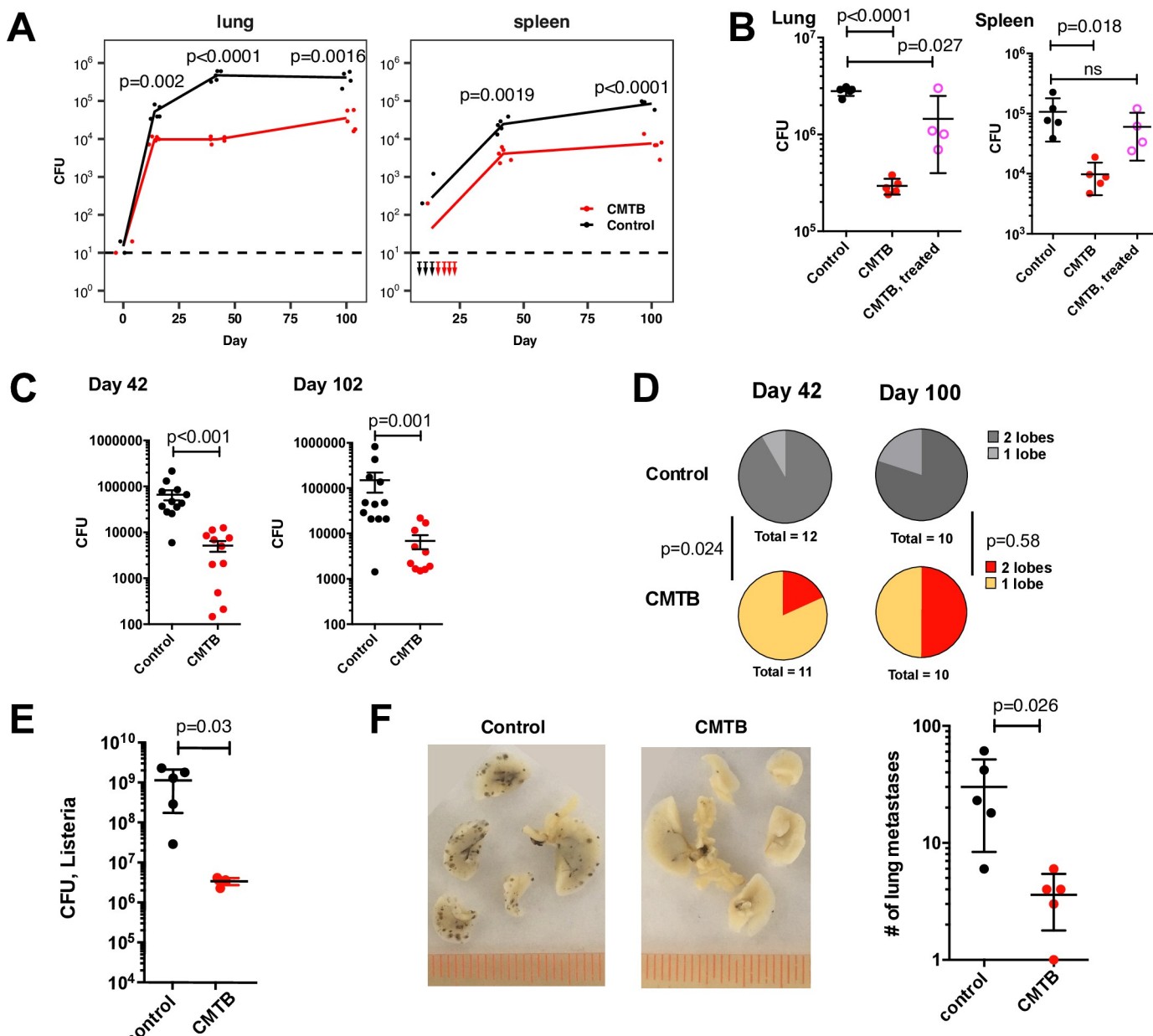

**Fig 2. Contained TB infection provides sustained protection against aerosol challenge with Mtb and heterologous challenges.** (**A**) 8 weeks after the establishment of CMTB, mice were challenged with 50–100 CFU of Kanamycin-resistant H37Rv via aerosol. Bacterial burdens in the lungs and spleen were measured by CFU at days 14, 42 and 100 following aerosol infection using plates containing Kanamycin. (Mtb was never detected in the spleen of CMTB mice at day 14.) Plot shows representative data from one of two independent experiments (n = 4–5 mice per group). Statistical significance was determined by Student's *t*-test. (**B**) CMTB was established and after 2 weeks CMTB and control mice were treated for 6 weeks with Isoniazid and Rifampicin. Mice were challenged via aerosol with 50–100 CFU of H37Rv Mtb and bacterial burden in the lung and spleen measured at 6 weeks. Data are representative of 2 independent experiments with 4–5 mice per group. Error bars depict mean and SD. Statistical significance was determined by Student's *t*-test. (**C**) CMTB and control mice (20 per group) were infected with an average of 1 CFU of Mtb H37Rv via aerosol. Bacterial burdens in the lung were measured by CFU. Data are representative of two independent experiments. Mice with no detectable bacteria were omitted from the plot. Statistical significance was determined using the Mann-Whitney test. Error bars depict the mean and SEM. (**D**) Fractions of control and CMTB mice with detectable bacteria in one or two lobes of the lung in the experiments shown in (**C**). Uninfected mice are excluded from the plot. Significance was assessed using the exact test for the difference of means in two Poisson distributions. (**E**) CMTB and control mice were challenged i.v. with $10^5$ CFU of *Listeria monocytogenes* and bacterial burden in the spleen measured 48 hours following infection by CFU. Statistical significance was determined by Student's *t*-test. Data are representative of two independent experiments with 4–5 mice per condition. (**F**) CMTB and control mice were challenged i.v. with 1x$10^6$ B16-F10 melanoma cells. Disease burden was quantified by counting the number of metastases visible in the lung 10 days following challenge (left panel, black spots). Data are representative data of three independent experiments with 4–5 mice per group. Significance was assessed by Student's *t*-Test. Error bars depict the mean and SD.

delivery of $1\times10^6$ CFU) which induces 10-fold protection at day 42 that wanes by day 100 and has minimal impact on bacterial burden during the first two weeks (S4 Fig).

We hypothesized that the protective effect of CMTB results in part from the continuous interaction between the immune system and live bacteria and would be diminished by antibiotic treatment. Therefore, we established CMTB and after 2 weeks treated mice for 6 weeks with Isoniazid and Rifampicin. Treatment efficacy was confirmed by culture of lymph node, spleen, and lung lysates from a dedicated set of mice (S5 Fig). Mice were then challenged via aerosol with 50–100 CFU of H37Rv Mtb and bacterial burden in the lung and spleen was measured at 6 weeks. Antibiotic treatment strongly diminished the ability of CMTB mice to control bacterial growth in the lung and spleen over the first 6 weeks following aerosol infection (Fig 2B).

Several lines of evidence suggest that human TB typically arises from infection with as few as 1–3 bacteria. We therefore examined the protective effect of CMTB against an "ultra-low dose" (ULD) challenge with a targeted average dose of ~1 CFU per mouse (see [27], and manuscript in review). At this dose, 30–50% of mice remain uninfected and infected mice exhibit a wide range of outcomes, especially at later times, with bacterial burdens in the lung ranging over nearly 4 orders of magnitude (Fig 2C). Although CMTB did not affect the likelihood of becoming infected (56.4% (22/39) of control mice infected compared to 52.5% (21/40) of CMTB mice infected, see Methods), for those mice that were infected the average bacterial burden in CMTB mice was more than 10-fold lower than in infected control mice (Fig 2C), consistent with the 50–100 CFU aerosol infections. Furthermore, because most mice are infected with a single bacterium, the degree of dissemination can be inferred by measuring spread to the contralateral lung lobe. CMTB mice exhibited reduced dissemination in the lung compared to controls at day 42 following ULD aerosol challenge (Fig 2D). These data demonstrate that CMTB mice are strongly protected against aerosol challenge with Mtb from as early as day 14 until at least 100 days following infection.

Both innate and adaptive immune responses are required for control of Mtb. We reasoned that if the protective phenotype of CMTB mice is fully explained by the presence of Mtb-specific T cells (Fig 1C), these mice should be equally susceptible to other immune challenges that do not share specific antigens with Mtb. To test this, we challenged CMTB and control mice intravenously with $10^5$ CFU of the intracellular bacterial pathogen *Listeria monocytogenes*. CMTB mice were significantly protected, displaying a greater than 10-fold reduction in bacterial burden in the spleen 48 hours following infection (Fig 2E). To confirm and extend these observations to a non-bacterial challenge, we inoculated CMTB and control mice with B16-F10 melanoma cells which serve as a model of metastatic cancer. Ten days following i.v. injection of B16-F10 cells, CMTB mice had approximately 10-fold fewer metastases in the lungs than controls (Fig 2F). These results demonstrate that CMTB induces amplified innate immune responses and raises the possibility that these responses may contribute to protection against Mtb-infection, either directly, or through more robust recruitment of adaptive immunity.

## The protective phenotype of CMTB mice is associated with modest localized immune activation at baseline and an accelerated immune response

Based on the heterologous protection induced by CMTB, we hypothesized that it modulates the innate immune system at baseline and in response to subsequent challenge leading to greater restriction of bacterial growth.

Although CMTB induced only modest increases in peripheral blood cytokines, we hypothesized that it might drive low-level, chronic immune activation in specific tissues. Therefore, we measured cytokine levels in the lungs and spleens of CMTB and control mice prior to aerosol challenge. During the first 6 weeks after the establishment of CMTB, the levels of 8 cytokines and chemokines (of 38 assayed) were significantly (FDR < 0.05) elevated in the lungs of CMTB mice compared to controls and 6 were elevated in the spleen (Fig 3A). The most pronounced changes were increased levels of CCL3, CCL4, and CCL5 in the lung and IFNG and CCL4 in the spleen (Fig 3A). Relative levels of the detectable cytokines correlated well across tissues (S6 Fig). These results suggest that the localized, contained Mtb-infection induces a mild and selective increase in local tissue immune activation.

Although the composition of the myeloid cell populations in the lungs of CMTB and control mice were not significantly different (S7 Fig), we hypothesized that their function might be affected by elevated cytokine levels in CMTB mice. Therefore, we assessed the overall responsiveness of myeloid cell populations in the lung by isolating CD11b$^+$ cells from CMTB and control mice and stimulating them for 6 hours with TLR-agonists. Myeloid cells from CMTB mice were more responsive to all stimuli tested as measured by TNF-α secretion (Fig 3B).

Next, we investigated the immune response after aerosol Mtb-infection (50–100 CFU) by histology of fixed lung sections. In contrast to control mice, which had essentially no histologically detectable immune response to aerosol Mtb-challenge at day 14 following infection, CMTB mice exhibited well-formed pulmonary lesions that remained stable for at least 100 days (Fig 3C). By day 42, control mice showed significantly more lesions and pulmonary hemorrhages than CMTB mice and this damage continued to progress through day 100 (Fig 3C, S1 Table).

To identify the myeloid cells involved in the early response that we observed histologically, we measured the recruitment of immune cells to the lung by flow cytometry at early timepoints (days 10–14) after aerosol infection. More monocyte derived macrophages and Mtb-specific CD8 T cells were recruited to the lung at this early timepoint in CMTB mice compared to controls (Fig 3D–3E). Although the numbers of alveolar macrophages (AMs) did not differ between control and CMTB mice at 10 days following aerosol challenge (S8 Fig), AMs from CMTB mice were more highly activated as measured by MHC II expression (Fig 3F). In contrast, MHCII expression was equivalent for pulmonary monocytes extracted from control or CMTB mice (S9 Fig). While we were unable to detect recruitment of ESAT-6 specific CD4 T cells, we measured an increase in the overall number of CD4 T cells in the parenchyma (S10A Fig). ESAT-6 specific CD4 T cells were only detectable at week 6 after aerosol infection without differences between the groups (S10B Fig).

Antibiotic treatment to clear the contained infection reversed the accelerated immune response, including a reduction in 1) CD8 T-cell recruitment to the parenchyma (S11A Fig), 2) the expansion of Mtb-specific T cells in response to infection (S11B Fig), and 3) MHC II expression on AMs (S12 Fig).

These results suggest that a key feature of CMTB is an accelerated, early immune response that recruits effector cells including T cells and bone marrow derived macrophages reducing the time during which Mtb is able to replicate unchallenged. Furthermore, CMTB mice are able to maintain a protective response with minimal progression of immune pathology for a prolonged period of time. Finally, the early immune response observed in CMTB mice is strongly dependent on live mycobacteria.

## Contained Mtb infection alters the response of alveolar macrophages to aerosol Mtb challenge

Since the protective phenotype in CMTB mice is strongly associated with early recruitment of inflammatory cells to the site of infection (Fig 3) and there is evidence of low-level tissue

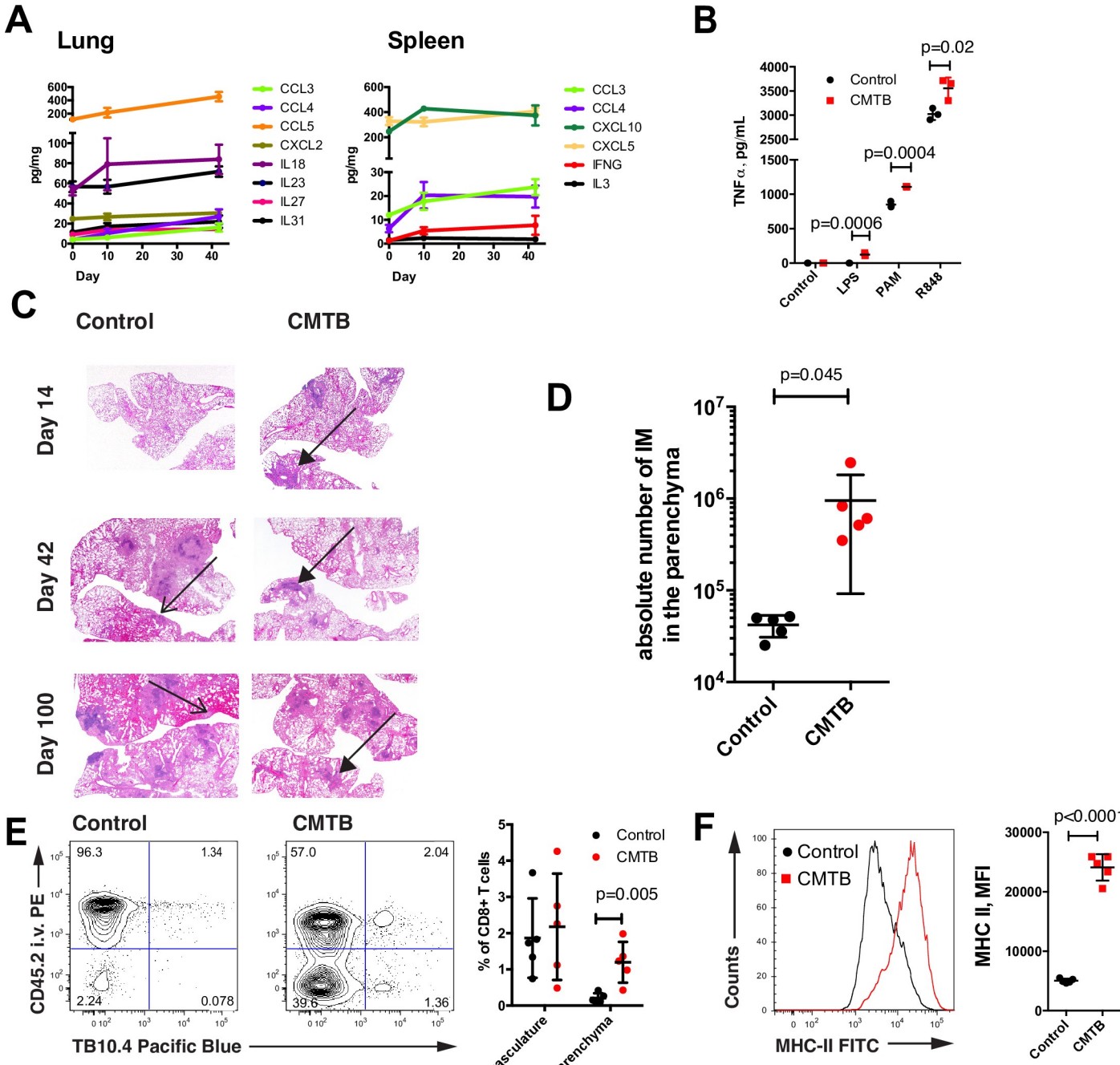

**Fig 3. The protective phenotype in CMTB mice is associated with elevated tissue inflammation and an early immune response.** (**A**) Tissues were isolated 10 or 42 days following the establishment of CMTB or from control mice (plotted at day 0). Absolute levels of cytokines and chemokines significantly altered by CMTB during at least one time-point in the lung or spleen (of 38 assayed) are plotted. Cytokine/chemokine amounts were normalized to total protein (*n* = 5 mice per time-point). A multiple *t*-test approach with Benjamini-Hochberg correction was used to test for significance (p < 0.05 for all analytes shown). (**B**) CD11b+ cells isolated from the lungs of CMTB or control mice were stimulated for 6 hours with the indicated TLR agonists and secreted TNF measured by ELISA (LPS 10 ng/mL, PAM3 300 ng/mL, R848 100 ng/mL). Data are representative of three independent experiments with cells from 3–5 mice pooled per condition. Points indicate technical replicates and significance was assessed by Student's *t*-Test. Error bars depict the mean and SD. (**C**) Haemotoxylin and eosin stained lung sections from control and CMTB mice at the indicated time-points following aerosol challenge. Regions of infiltration by immune cells (filled arrow heads) and intrapulmonary hemorrhages (open arrow heads) are indicated. The corresponding quantitative pathology assessments are presented in S1 Table. Representative images from 5 mice per condition are displayed at 2×. (**D**) Quantification of relative number of CD11b⁺CD11c⁺CD64⁺Siglec-F⁻ monocyte derived macrophages in the lung parenchyma in control and CMTB mice at 14 days following aerosol infection with 100 CFU of Mtb H37Rv. Data are representative of two independent experiments with mice (n = 4–5) per condition. Statistical significance was determined by Student's *t*-Test. Error bars depict the mean with SD. (See S19 Fig for gating strategy).). (**E**) Representative cytometry plot and corresponding quantification of CD3⁺CD8⁺TB10.4⁺ cells in the vasculature (CD45.2-PE⁺) and lung parenchyma (CD45.2-PE⁻) in control and

CMTB mice at 14 days following aerosol infection with 100 CFU of Mtb H37Rv. Statistical significance was determined by Student's *t*-Test. Error bars depict the mean with SD. (**F**) Expression of MHC II on alveolar macrophages (CD11b$^{int}$CD11c$^+$CD64$^+$Siglec-F$^+$) from control and CMTB mice isolated from the lung at 10 days following aerosol challenge. (See S19 Fig for gating.) Data are representative of two independent experiments with 5 mice per condition. Statistical significance was determined by Student's *t*-Test. Error bars depict the mean with SD.

inflammation prior to challenge (Fig 3A), we hypothesized that CMTB affects the activation status and initial response of AMs (the lung resident macrophage and the first cells to be infected with Mtb).

While the total number of CD45$^+$ cells and T cells as well as the fractions of various myeloid cell types in the lung were unaffected by CMTB (S7 Fig), we consistently measured elevated expression of MHC II and FcγRI (CD64) on AMs (Fig 4A), suggesting that their activation state was altered by continuous exposure to low-level inflammation. To investigate the global transcriptional response of these cells, we performed RNA sequencing on AMs isolated from CMTB and control mice. Interestingly, the transcriptomes of AMs from control and CMTB mice did not differ substantially prior to aerosol challenge (Fig 4B).

To assess the immediate response of AMs to infection with Mtb, we challenged mice with a high dose (~3000 CFU) of mEmerald-expressing bacteria. In concordance with previous studies [28], at 24 hours following infection the bacteria were predominantly contained within AMs in both control and CMTB mice (S13 Fig). The fraction of infected AMs was not altered by CMTB (S13 Fig). In other studies, we have shown that Mtb-infected AMs from control mice display a highly restrained response over the first week following infection and most pro-inflammatory genes are not expressed until approximately day 10 [28]. RNA-seq analysis of AMs isolated from control and CMTB mice 24 hours after aerosol challenge demonstrated that CMTB dramatically alters the AM response to Mtb infection (Fig 4C and 4D). Gene Set Enrichment Analysis (GSEA)[29] of genes differentially expressed following infection showed that, in contrast to AMs from control mice, AMs from CMTB mice upregulated transcripts associated with inflammation and with the Interferon-γ and Interferon-α pathways (Fig 4E and 4F).

The bacterial burden in CMTB mice at 10 days following high-dose (~3000 CFU) aerosol infection, was approximately 3-fold lower than in control mice (S14 Fig). The majority of bacteria were in AMs and significantly fewer AMs were infected in CMTB mice (Fig 4G). We also measured a corresponding increase in the numbers of infected polymorphonuclear leukocytes (PMNs) in CMTB mice, suggesting an accelerated cellular response compared to controls (Fig 4G). AMs isolated by BAL from CMTB mice were better able to control Mtb infection ex vivo compared to controls suggesting that the enhanced response of AMs from CMTB mice is at least in part cell-intrinsic (Fig 4H).

Numerous recent studies have suggested that prior or ongoing infection can alter the capacity of innate immune cells to respond to subsequent encounters with pathogens, a phenomenon that has been termed "trained immunity" and has been shown in some cases to be reflected in epigenetic modifications [30]. Therefore, we used ATAC-seq to measure genome-wide changes in chromatin accessibility induced by CMTB. Surprisingly, we observed only modest changes in chromatin accessibility that did not correlate with differentially expressed genes, suggesting that this mechanism cannot completely account for the enhanced response of AMs to infection (S15 Fig).

## The elevated immune activation in CMTB mice is dependent on systemic interferon-γ

We analyzed the activation state of mature monocytes in the peripheral blood of CMTB mice and found that a larger fraction expressed MHC II compared to control mice (Fig 5A). Given

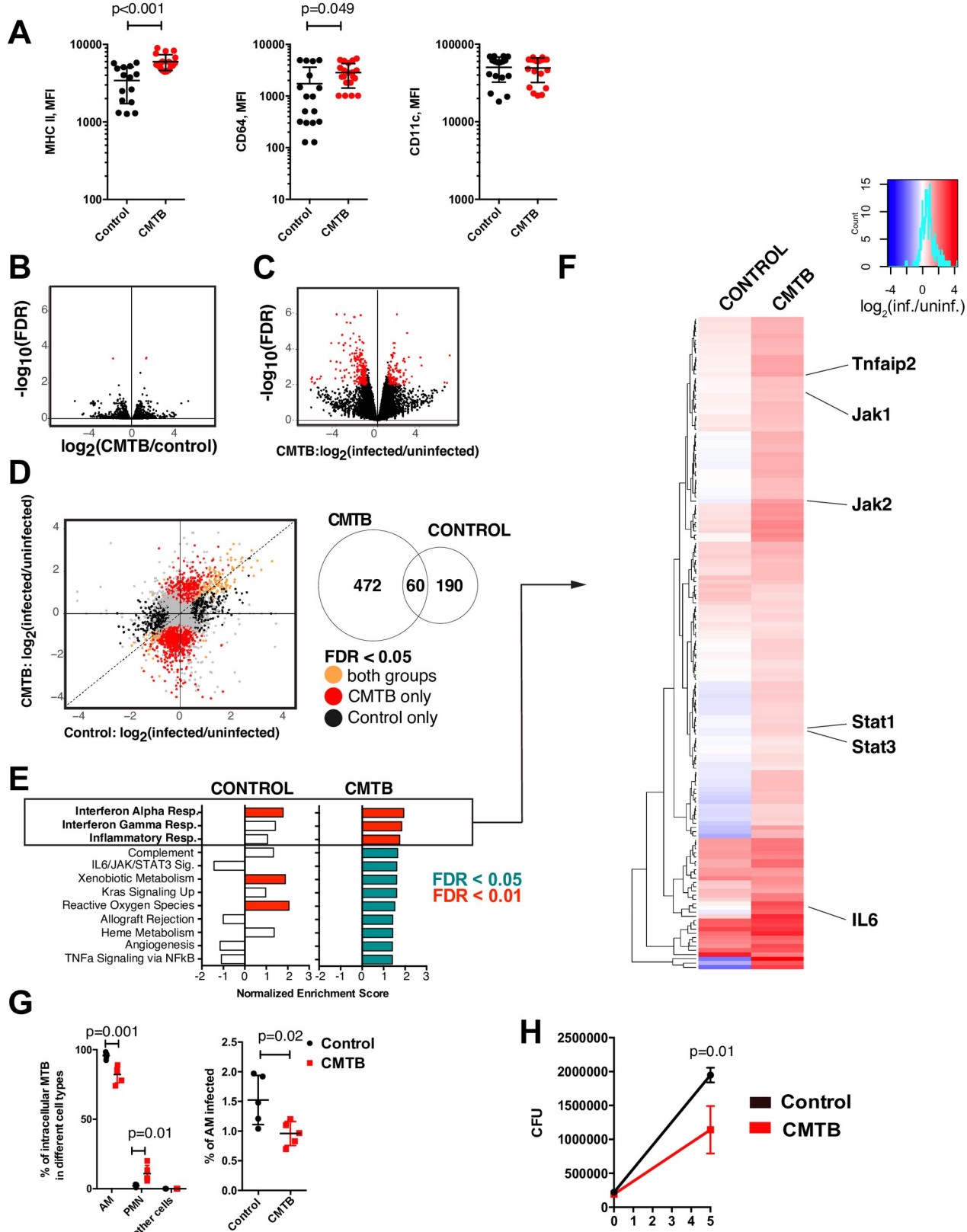

**Fig 4. Alveolar macrophages are reprogrammed by CMTB.** (**A**) Flow cytometry analysis of AMs from control and CMTB mice (See S19 Fig for gating). Expression of the indicated markers on AMs was quantified by MFI. Significance was determined by Student's *t*-Test. Error bars depict the

mean and SEM. (Data are pooled from 4 independent experiments with 3–5 mice per experiment). (**B**) AMs from CMTB mice and controls were isolated by FACS and the transcriptome assessed by RNA sequencing. The plot shows significance vs. fold-change in transcript expression between CMTB and control AMs. Red dots represent transcripts that were significantly differentially expressed between conditions ($|\log_2(\text{fold-change})| > 1$ and FDR < 0.05). Each point represents the average of 3 biological replicates each of which consists of BAL pooled from 10 mice. (**C**) Infected and uninfected AMs from CMTB mice and controls were isolated by FACS 24 hours after aerosol challenge with ~3000 CFU of mEmerald-expressing Mtb. Red dots represent transcripts that were significantly differentially expressed between conditions ($\log_2(\text{fold-change}) > 1$ and FDR < 0.05). Each dot represents the average of two biological replicates, each of which consists of BAL pooled from 10 mice. (**D**) (left panel) Transcripts differentially expressed (as defined in C) in AMs from control mice compared to AMs from CMTB mice. Transcripts responding to infection are color-coded by whether they are differentially expressed in: red = CMTB only, black = control only, orange = both conditions. (right panel) VENN diagram showing the number of differentially expressed transcripts in each group. (**E**) Enrichment scores for the 10 most enriched pathways in CMTB AMs defined by GSEA of transcripts altered by Mtb infection at 24 hours in AMs from control and CMTB mice are shown. Red (green) bars indicate significant enrichment with FDR < 0.01 (0.05). White bars indicate FDR > 0.05. (**F**) Heatmap showing the fold changes of unique, significantly enriched leading edge genes from the GSEA. Key transcripts are highlighted. (**G**) Control and CMTB mice were infected with ~3000 CFU of mEmerald-expressing Mtb and the distribution of infected cell types determined by flow cytometry at 10 days following challenge. *Left panel*: Quantification of the distribution of infection across cell types. *Right panel*: Fraction of AMs infected. (See S19 Fig). Data are representative of three replicate experiments with 4–5 mice per condition. Significance was determined by Student's *t*-Test. Error bars depict the mean and SD. (**H**) AMs from CMTB mice and controls were harvested by bronchoalveolar lavage and infected with H37Rv *ex vivo* at an MOI of 1. Bacterial burden was measured by CFU at 2 hours and 5 days following infection. Data are representative of two independent experiments with 4–5 mice per condition. Significance was measured by Student's *t*-test. Error bars depict the mean and SD.

that the enrichment analyses of transcriptomes from AMs in CMTB mice suggested a prominent role of IFNG signaling and that IFNG plays a significant role in the control of both *Listeria* infections and melanoma [31,32], we hypothesized that low grade peripheral IFNG cytokinemia might be the underlying cause of the elevated activation state. To test this, we established CMTB in WT:*Ifngr1*⁻ᐟ⁻ mixed bone marrow chimeras. In concordance with this hypothesis, MHC II was upregulated by CMTB in wild-type circulating monocytes but not in monocytes lacking Ifngr1 (Fig 5B). The activation state of AMs after aerosol Mtb challenge was similarly dependent on Ifngr1 with wild-type cells exhibiting elevated MHC II in CMTB that was not present in *Ifngr1*⁻ᐟ⁻ cells (Fig 5C). Taken together, these data suggest that a significant portion of the altered activation state of both circulating and lung-resident innate cells in CMTB mice arises from low levels of systemic IFNG.

## Discussion

For years, efforts to develop an effective TB vaccine have been stymied because we do not completely understand the nature of protective immunity that must be induced. Several inherent limitations restrict the ability to uncover the correlates of immune protection solely through human investigations: 1) Although natural immunity exists, it is difficult to definitively identify those who are protected. 2) Protective immunity is probably established in the first days to few weeks after aerosol Mtb exposure, but infected individuals are usually not identified until many weeks or months later. 3) Protective immune responses occur at tissue sites of infection which are generally inaccessible. 4) While correlates of protection can be identified, in vivo mechanistic dissection cannot be performed. Thus, the mouse continues to play an essential role as a model organism for the development of TB vaccine strategies due to the extensive availability of molecular reagents, a large body of historical data, and relatively low cost [33]. Much of our current understanding of TB immunology had its origins in the mouse before subsequent validation in humans, for example the critical roles for CD4 T cells [34], IFNγ [35,36], IL-12 [37], and TNF [38]. Despite this intensive research focus, the cellular and molecular correlates of a protective vaccine remain poorly defined. In this study, we have demonstrated that contained Mtb-infection (CMTB) protects mice against subsequent aerosol challenge. In contrast to BCG, the "gold standard" and most widely studied vaccine in mice, CMTB leads to a reduced bacterial burden as early as 10 days following aerosol challenge which is maintained for at least 3 months. To our knowledge, this protection is as rapidly

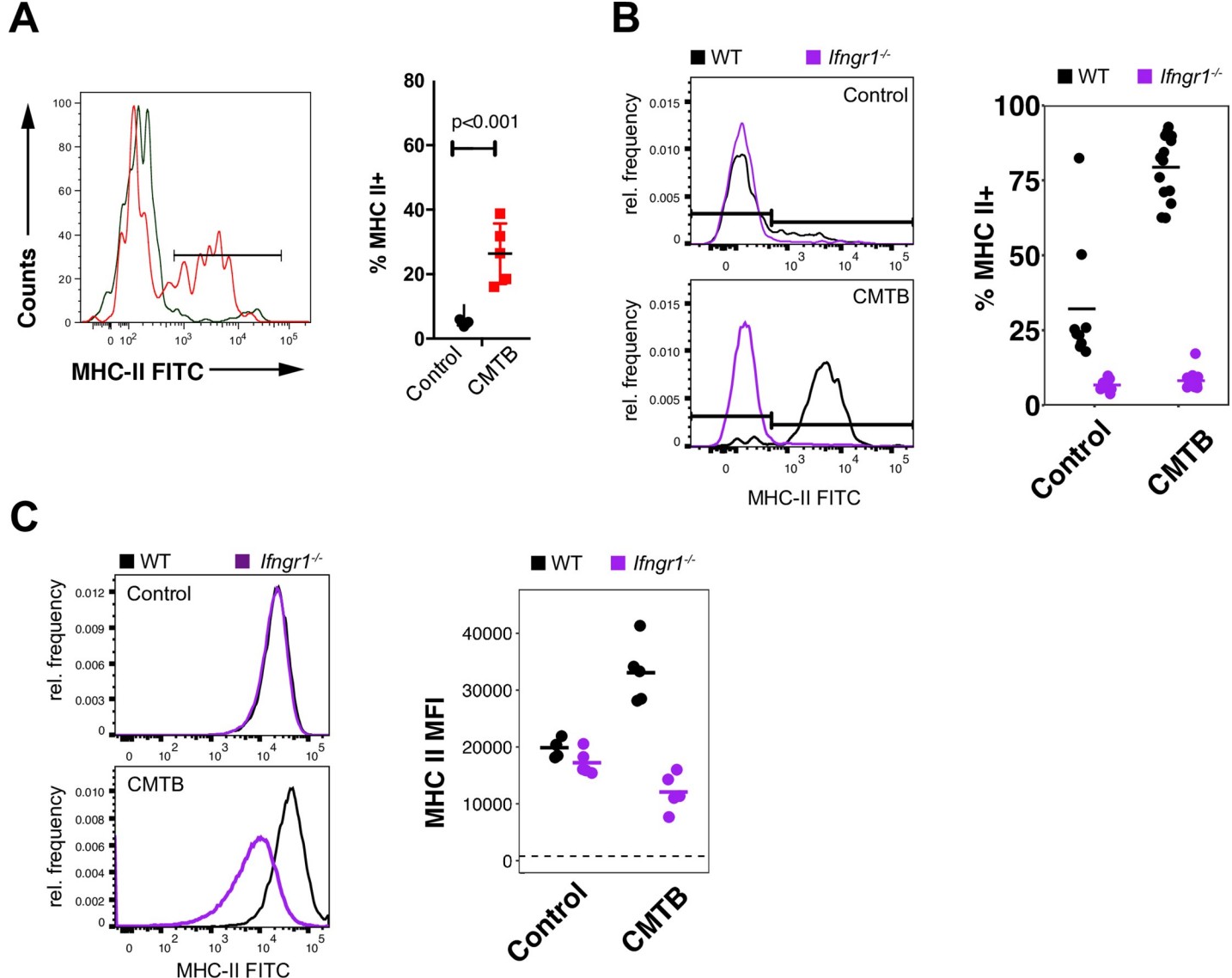

**Fig 5. Immune activation in CMTB mice depends on systemic IFNg signaling.** (**A**) MHC II expression on circulating monocytes in control and CMTB mice 6 weeks following the establishment of CMTB. (See S17 Fig for gating strategy.) Significance was measured by Student's *t*-test. Error bars depict the mean and SD. (**B**) MHC II expression on circulating monocytes in WT:*Ifngr1*-/- mixed bone marrow chimeras with and without CMTB. (See S20 Fig for gating strategy.) (**C**) MHC II expression on alveolar macrophages in WT:*Ifngr1*-/- mixed bone marrow chimeras with and without CMTB. Dashed line indicates MFI of MHC II in unlabeled cells. (See S21 Fig for gating strategy).

acting and durable as any that has been demonstrated in the mouse [33]. Both of these qualities are highly desirable in a candidate vaccine against adult TB. Therefore, we argue that the CMTB mouse model offers an important opportunity to identify and dissect mechanisms of protection against TB through detailed studies comparing CMTB and BCG in order to reveal the correlates that distinguish these protective modalities.

While primed T cell responses undoubtably play a significant role in the protective effect of CMTB, our data suggest that amplified AM responses also contribute. In control mice, tissue resident AMs exhibit a cell-protective antioxidant response and have been shown to be the predominant niche for Mtb growth over the first week following infection [28,39]. In contrast,

myeloid derived macrophages are better able to control bacterial replication. In fact, a recent study raised the possibility that the protection induced by BCG vaccination might be partially explained by an accelerated transfer of Mtb from AMs into bone-marrow-derived, inflammatory macrophages [40]. The in vivo transcriptional response of AMs from CMTB mice, as early as 24 hours following infection, is altered towards a proinflammatory phenotype; Mtb-infected AMs from CMTB mice upregulate many of the inflammatory and interferon pathways that are activated in myeloid-derived macrophages but not in Mtb-infected AMs from control mice [28]. In addition to this shift towards a protective transcriptional program, AMs from CMTB mice infected with Mtb ex vivo control bacterial growth better than those from control mice. In vivo, at 10 days following aerosol challenge, fewer AMs from CMTB mice are infected compared to controls, suggesting that bacterial spread is more effectively contained.

One outcome of a more robust innate response is an accelerated adaptive immune response. Our data suggest that systemic low-level inflammation changes the local lung environment in a manner that elevates the activation state of resident AMs, including MHC II expression, and that a substantial portion of this effect can be attributed to IFNG. Given that the total number of CD45$^+$ cells in the lung does not change, that we measure elevated levels of inflammatory mediators such as IFNG, IL6, and CXCL10 in the blood, and that we observe increased activation of circulating innate immune cells, we believe it is likely that the increase in tissue cytokine levels arises from low grade cytokinaemia. However, we cannot exclude the possibility of a contribution from immune cells that were activated in the cervical lymph node and subsequently migrated to the lung. Our data demonstrate that CMTB leads to a profound and early influx of both CD4+ and CD8+ T cells after aerosol infection, which likely contributes to the protective effect and the long-term maintenance of protective immunity. Because T cells are critical for maintaining CMTB [21], isolating the contributions of innate and adaptive immune mechanisms to protection is challenging. The IFNG which drives elevated activation of AMs in CMTB mice most plausibly arises from Mtb-specific T cells, and therefore the distinction between "innate" and "adaptive" mechanisms in this model is somewhat artificial. More detailed studies are required to define the contributions of AM-intrinsic bacterial control, enhanced antigen-presenting cell activity, and memory T cells to the protective effect of CMTB.

Although we easily detect circulating TB10.4-specific CD8+ T cells in CMTB mice prior to aerosol challenge we were unable to detect ESAT-6 specific CD4+T cells in any compartment. In other studies using C57BL/6 mice, we detected much greater numbers of TB10.4-specific CD8+ T cells than ESAT-6-specific CD4+ T cells in both the lung and draining lymph nodes during the first month following aerosol infection [41]. Given that the bacterial burden in lymph node in the CMTB model is much lower than that in the lung at 30 days following aerosol challenge ($10^4$ CFU vs. $10^5$–$10^6$ CFU) our results are consistent with previous measurements of the relative abundances of these T cell subsets induced by Mtb-infection.

Concomitant immunity, defined as a low-grade infection that protects against subsequent infections with the same pathogen, has been described for several different pathogens, but most commonly for helminths and protozoan parasites [42–44]. For example, in a mouse model of Leishmania infection, sterilizing protection against secondary challenge was dependent on persisting parasites from the first infection [43]. In this study we show that the protection afforded by CMTB is dependent on the persistence of live bacteria: Antibiotic treatment of CMTB mice reduces the numbers of Mtb-specific T cells in the circulation and in the lung parenchyma following aerosol challenge and eliminates their expansion following infection. Furthermore, antibiotic treatment partially reverses elevated MHC II expression on tissue-resident AMs.

Several lines of evidence indicate that prior infection with Mtb, and presumed ongoing asymptomatic containment, provides significant protection against reinfection in humans. Studies from the pre-antibiotic era suggest that nursing students who were previously infected with Mtb were less likely to develop TB disease in a high-exposure setting than those who were not [13]. On an epidemiological level, it is impossible to accurately model Mtb transmission in highly endemic regions without accounting for the protective effect of prior infection and increased susceptibility after treatment [14]. Since a sub-clinical, contained infection is assumed to be occurring in a subset of humans who have been exposed to Mtb [22,45] and post mortem studies suggest that lymph nodes are a significant reservoir of bacteria [3,7,9], we believe that CMTB in mice reflects a portion of the protective mechanisms of natural immunity against TB in humans and provides a needed model to better understand the underlying mechanisms.

The potential role of CMTB-mediated heterologous protection in humans has not been rigorously assessed, however, beneficial heterologous effects of BCG vaccination have been reported. Overall mortality of BCG-vaccinated newborns is reduced by more than would be expected from the protection afforded by BCG against TB alone [46] and the observation that BCG has profound, heterologous beneficial effects in both children and adults provides an additional argument in favor of BCG vaccination [46–49].

In most adults, infection with Mtb leads to a paucibacillary condition that is asymptomatic in ~95% of cases. In both contemporary and historical epidemiological studies, chronic paucibacillary or "latent" Mtb infection has been shown to confer significant protection against subsequent challenge. The vast majority of humans with latent TB acquire the bacteria by inhalation whereas CMTB in mice is established by intradermal inoculation, leaving the lung naïve to infection. Nevertheless, the CMTB model shares numerous features with latent tuberculosis in humans including live, contained bacteria, absence of overt symptoms, circulating Mtb-specific T cells, and a requirement for immune modulators such as TNF in maintaining bacterial containment. The current medical standard for patients with latent tuberculosis infection in non-endemic areas is prophylactic antibiotic treatment. Given the non-negligible side-effects of the current antibiotic regimes and the increasing prevalence of antibiotic-resistant Mtb, the appropriate management for latent tuberculosis in high-incidence settings remains a matter of considerable debate. While our work cannot provide a definitive answer to this public health question, it does contribute to our understanding of CMTB and raises important issues that need to be considered when crafting population-level strategies for TB control. A deeper understanding of the protective mechanisms generated by CMTB coupled with the development of related biomarkers to risk-stratify patients, would substantially inform clinical practice.

## Methods

### Ethics statement

All animal experiments were approved by the Institutional Animal Care and Use Committee at Seattle Children's Research Institute. Mice were sacrificed by $CO_2$ asphyxiation and/or cervical dislocation.

### Mice

C57BL/6 were purchased from the Jackson Laboratory (Bar Harbor, ME). Mice were housed and maintained in specific pathogen–free conditions at the Seattle Children's Research Institute (SCRI), and experiments were performed in compliance with the Institutional Animal Care and Use Committee. CMTB was established in 6- to 12-week-old male and female mice.

Mice infected with Mtb. were housed in a biosafety level 3 facility in an animal biohazard containment suite. Cages of genetically identical mice were randomly assigned to experimental groups.

## Tissue culture

Alveolar macrophages were isolated from bronchoalveolar lavage (BAL) fluid by plate adherence (12 hours) and cultured in complete RPMI [cRPMI; plus 10% (vol/vol) FBS, 2 mM L-glutamine, penicillin, and streptomycin] for 24 hours. In vitro infections were performed in cRPMI without antibiotics in biological triplicate (cells from independent mice). The bacterial load at day 5 was determined by plating serial dilutions of cells lysed in 1% Triton-X and diluted in 0.1% Tween-80 PBS.

## Establishment of CMTB via intradermal infection of the ear, aerosol infections, and quantification of bacterial load

Intradermal infections to establish CMTB were performed as described previously [21] with the following modifications: 10,000 CFU of Mtb (H37Rv) in logarithmic phase growth in 10 μL PBS were injected intradermally using a 10μL Hamilton Syringe into mice anaesthetized with Ketamine. In some experiments, Mtb Erdman was used with identical results. For aerosol infections, a frozen stock of Kanamycin-resistant Mtb H37Rv was diluted and used to infect mice in an aerosol infection chamber (Glas-Col). For high dose mEmerald infections, a deposition of 3000–5000 CFU was targeted. High dose aerosol infections were performed with a stock of wild-type H37Rv transformed with an mEmerald reporter pMV261 plasmid, generously provided by Dr. Chris Sassetti and Dr. Christina Baer (University of Massachusetts Medical School, Worcester, MA). For Ultra-low-dose (ULD) infections, a deposition of 1–3 CFU was targeted accepting a rate of approximately 30% uninfected animals. Bacterial load was determined independently for the left and right lungs by plating serial dilutions from homogenized tissue. Homogenates were plated on Kanamycin-containing plates and antibiotic-free plates to distinguish the origin of the infection (intradermal or aerosol infection). The percentage of mice infected (in one or more lungs) in each experimental condition was taken as a measure of the susceptibility to infection. This analysis assumes that no mice clear infecting bacteria to below the detection limit. The assumption of no clearance is supported by the observations that for each experimental condition (control or CMTB) there was no difference in the fraction of mice infected between day 42 and day 102 and that the average bacterial load did not decrease over this interval (Fig 2C). The degree of dissemination between lungs in each experimental group was estimated by assuming that dissemination is a Poisson process. For the actual infection rates achieved in the ULD experiments (~50% infected) 1 in 10 mice would be expected to be infected with more than 1 bacterium. In determining the statistical significance of a difference in the frequency of dissemination events between conditions, we make the conservative assumption of ignoring multiple infections (Fig 2D). For Mtb-infection experiments, the group sizes and experimental replication strategies were chosen based on previous experience with the "high-dose" (2000–4000 CFU), "conventional-dose" (CFU), and "ultra-low dose" (1–3 CFU) infection models to ensure robust detection of protective effects as large as those observed for BCG vaccination of mice.

## Antibiotic treatment to clear CMTB

CMTB was established and after 2 weeks mice were treated for 6 weeks with Isoniazid (0.1 mg/mL) and Rifampicin (0.1 mg/mL) in the drinking water. After the end of treatment, mice were switched to untreated water for at least 1 week to allow complete removal of the antibiotics

from the system (Isoniazid half-life: 4 hours; Rifampicin half-life RIF: 2.5 hours). Efficacy of treatment was confirmed by plating undiluted tissue homogenates of cervical lymph nodes, lungs, and spleens.

## Heterologous challenges

For experiments with bacterial burden as the endpoint, $10^5$ Listeria monocytogenes were injected i.v. and CFU in the spleen measured after 48 hours. For experiments with macrophage activation as the endpoint, mice were infected with $10^7$ Listeria monocytogenes and sacrificed 2 hours after i.v. infection. The B16-F10 melanoma cell line (ATCC CRL-6475) was purchased from ATCC and expanded according to the supplier's instructions. For the melanoma challenges, $10^5$ melanoma cells per mouse were injected i.v. After 10 days, mice were sacrificed, lungs were extracted and bleached in Fekete's solution following a published protocol [50], and the metastases were counted by an investigator blinded to the identity of each mouse.

## Cell isolation, analysis, and sorting

For i.v. labeling, a PE labeled anti-CD45.2 antibody was injected i.v. 10 minutes prior to tissue harvest. Single-cell suspensions of lung cells were prepared by Liberase Blendzyme 3 (Roche) digestion of perfused lungs as previously described [51]. Cells from spleens were prepared as previously described (Spleen digestion protocol, Miltenyi Biotec). Fc-receptors were blocked with an anti-CD16/32 antibody (clone 2.4G2). Cells were suspended in PBS (pH 7.4) containing 2.5% FBS and stained in saturating conditions with antibodies against various epitopes (S2 Table). Live cells were identified using Zombie Violet or Zombie Aqua (BioLegend). Absolute numbers of cells were estimated using CountBright Absolute Counting Beads (Invitrogen) according to the published protocol of the manufacturer. Samples were fixed in 2% (vol/vol) paraformaldehyde and analyzed using a LSRII flow cytometer (BD) and FlowJo software (Tree Star, Inc.). Previously published gating strategies were followed [52,53] and are shown in S16–S21 Figs. For RNA-seq analyses, live alveolar macrophages were isolated from suspensions of lung cells using a BD Aria II cell sorter. Gating strategies are presented in S16 Fig (T cells), S17 Fig (Circulating monocytes), S18 Fig (Splenic myeloid cells), S19 Fig (Lung myeloid cells), S20 Fig (Peripheral blood monocytes in WT/*Ifngr1*$^{-/-}$ mixed bone-marrow chimeras), and S21 Fig (Alveolar macrophages in WT/*Ifngr1*$^{-/-}$ mixed bone-marrow chimeras).

## CD11b$^+$ cell enrichment and ex vivo stimulation

Single cell suspensions prepared as described above from pooled lungs or spleens were positively enriched for CD11b$^+$ using magnetic beads (Miltenyi Biotec). The cells were re-stimulated with TLR-agonists (LPS 10 ng/mL, PAM3 300 ng/mL, R848 100 ng/mL) and supernatants were collected 6 hours after re-stimulation and assayed by ELISA for TNFα.

## Measurement of cytokines and total protein

Cytokines were measured by the Cytokine & Chemokine 36-plex Mouse ProcartaPlex panel (with a LEP and IFNB1 assays added, for a total of 38 analytes) using a Luminex Bio-Plex 200 analyzer per the manufacturer's instructions. For spleen and lung tissue, total protein levels were normalized using a commercial BCA assay from Thermo Scientific.

## Detection of Mtb-specific T cells

For direct detection of Mtb-specific cells, APC-labeled MHC class II tetramers (I-Ab) containing the immunodominant epitope of the ESAT-6 protein of Mtb (ESAT-64–17:I-Ab) were

made in house [54]. Pacific Blue-labeled MHC class I tetramers containing the stimulatory residues of the TB10.4 protein of Mtb (TB10.4 4–11:Kb) were obtained from the National Institutes of Health Tetramer Core Facility. Tetramer staining on single-cell preparations was carried out as described previously [54].

## Histology

Dissected mouse lungs (left lobe) were fixed in 10% neutral-buffered formalin, processed routinely into paraffin, and stained with hematoxylin and eosin (H&E). H&E slides were digitized with an Olympus Nanozoomer and images captured with Nikon Digital Pathology viewing software. Samples were scored on a 0–4 severity scale (0 = normal or none; 1 = minimal; 2 = mild; 3 = moderate; 4 = severe) for the levels of alveolar hyperplasia, necrosis, and edema. Samples were scored on a 0–4 extent scale (0 = normal; 1 = <5%; 2 = 6–30%, 3 = 31–60%, 4 = >60%) for the fraction of the lung affected in any manner (Extent 1) and for the fraction of the lung affected in the worst manner (Extent 2). Samples were scored on a 0–4 scale (0 = none; 1 = few (<5 in focus); 2 = mild numbers (5–10); 3 = moderate numbers (11–20); 4 = marked numbers (>20)) for numbers of mixed granulomas (ill-formed granulomas with a mixture of macrophages and lymphocytes), defined granulomas (well-defined with separation of macrophages, epithelioid, or multinucleated giant cells with lymphoid aggregates), perivascular lymphoid aggregates, peribronchiolar lymphoid aggregates, histocytes (macrophages), foamy macrophages, neutrophils, cholesterol clefts, and acid-fast bacteria. Scoring was performed by a pathologist who was blinded to the experimental condition of each sample. For each condition, tissues from 5 mice were examined to determine a score for each parameter. An overall score was assigned for each condition by summing the scores for each parameter.

## Mixed-bone marrow chimeric mice

Bone marrow cells were harvested from femurs and tibias. T cells were depleted from bone marrow cell suspensions with an anti-CD3e microbead kit (Miltenyi). CD45.1/2-expressing WT bone marrow (from C57BL/6 × B6.SJL-Ptprc$^a$; Jax Stock No. 002014) cells were mixed with an equal number of CD45.2-expressing *Ifngr1*$^{-/-}$ (B6.129S7-*Ifngr1*$^{tm1Agt}$/J; Jax Stock No. 003288). We injected 5–10×10$^6$ total bone marrow cells into sub-lethally irradiated (600 rads) CD45.1-expressing mice (B6.SJL-Ptprc$^a$; Jax Stock No. 002014). Mice were allowed to reconstitute for 3 months, were bled, and circulating T cells were stained for CD45.1/CD45.2 to assess chimerism before infection.

## RNA-seq

RNA isolation was performed using TRIzol (Invitrogen), two sequential chloroform extractions, Glycoblue carrier (Thermo Fisher), isopropanol precipitation, and washes with 75% ethanol. RNA was quantified with the Bioanalyzer RNA 6000 Pico Kit (Agilent). cDNA libraries for alveolar macrophages were constructed and amplified using the SMARTer Stranded Total RNA-Seq Kit v2—Pico Input Mammalian (Clontech) per the manufacturer's instructions. Libraries were amplified and then sequenced on an Illumina NextSeq (2 x 75, paired-end). Stranded paired-end reads of length 76 were preprocessed: For the Pico Input prep, the first three nucleotides of R2 (v2 kit) were removed as described in the SMARTer Stranded Total RNA-Seq Kit—Pico Input Mammalian User Manual (v2: 063017); Read ends consisting of 50 or more of the same nucleotide were removed. The remaining read pairs were aligned to the mouse genome (mm10) + Mtb H37Rv genome using the GSNAP aligner (v. 2016-08-24) allowing for novel splicing. Concordantly mapping read pairs (average 10–20 million / sample) that aligned uniquely were assigned to exons using the subRead (v. 1.4.6.p4) program and gene

definitions from Ensembl Mus_Musculus GRCm38.78 coding and non-coding genes. Differential expression was calculated using the edgeR package from bioconductor.org. False discovery rate was computed with the Benjamini-Hochberg algorithm. Raw and processed data are deposited in GEO (GSE126355).

## ATAC-seq

The ATAC-seq protocol with modifications for PFA fixed cells as described by Chen et al. (2016) was used [55]. Libraries were sequenced on a NextSeq 500 (Illumina) using a 150 (paired-end 2x76bp) cycle mid-output kit. Unique sequence read pairs were aligned to the combined *Mus musculus* (mm10) and *Mycobacterium tuberculosis* (H37Rv) genomes using the GSNAP aligner [56,57] (v. 2016-08-24) after stripping off adapter sequences in a pairwise manner. Only pairs that aligned uniquely and concordantly to non-mitochondrial mouse chromosomes were retained. Start and end positions of the sequences were adjusted to extend 4 and 5 base pairs respectively to account for transposase adapter insertion (see Buenrostro et al. (2013) [58]). Peak calling was performed with MACS2 (2.1.0) [59] using the start and end locations of the pairs to define fragment lengths. "Blacklisted" regions of known artificially high signal as defined by the ENCODE project were filtered out of peak regions (https://sites.google.com/site/anshulkundaje/projects/blacklists) [60]. Bigwig files for each biological group were generated by running MACS2 [59] (https://github.com/taoliu/MACS) peak calling on combined alignments from all samples in the group and outputting a normalized bedgraph file followed by file conversion using the bedGraphToBigWig program (genome.ucsc.edu). The R package DiffBind [61] was used to define consensus peak regions across samples and assign counts. Differential peak counts and significance were computed using the R package edgeR [62,63]. Raw and processed data are deposited in GEO (GSE126355).

## Statistical analysis

Significance was determined using an unpaired two-tailed Student's *t*-test unless otherwise specified.

## Supporting information

**S1 Fig. Dissemination of Mtb to the spleen in the CMTB model.** Mice were inoculated intradermally in the ear with 10,000 CFU of H37Rv and the bacterial burdens in the spleen and lung were measured at 10 days, 6 weeks, and 1 year by CFU assay (4–5 mice/timepoint). No bacteria were detected in the lung in any sample (detection limit 10 CFU / lung).
(PDF)

**S2 Fig. Serum cytokines in CMTB mice.** Blood was drawn from mice prior to or at 10 or 42 days following the establishment of CMTB and the serum concentrations of the indicated proteins measured by multiplexed immunoassay. Absolute levels of cytokines and chemokines normalized to the total protein in each sample are plotted. Horizontal lines indicate the mean values of measurements above the quantification limit. The quantification limit for each analyte is indicated with a dashed line. Measurements below the quantification limit are plotted with red markers at half of that value and measurements below the detection level are plotted with open markers. (*n* = 5 mice per condition)
(PDF)

**S3 Fig. Protective effect of CMTB assessed at 6 weeks following aerosol challenge.** CMTB was established as described in the main text and mice challenged with 50–100 CFU of Mtb H37Rv via aerosol after 10–14 days ("Early", 2 replicates) or after 8–10 weeks ("Late", 4

replicates). Bacterial burden in the lung was measured by CFU assay. CMTB mice had on average 18.4-fold (CI: 10.6–26.3) fewer bacteria in the lung as compared to controls. In each individual experiment, the bacterial burden in CMTB mice was lower than that in control mice as determined by Student's *t*-test (*p* < 0.05). Error bars depict mean and SEM. (n = 3–5 mice per group).
(PDF)

**S4 Fig. Protective efficacy of BCG Pasteur.** Mice were immunized sub-cutaneously with 1x10$^6$ CFU BCG Pasteur and challenged with 100 CFU Mtb H37Rv via aerosol after 2 months. Bacterial burden in the lung was measured by CFU assay at days 10, 42, and 100 following aerosol challenge (n = 4–5 mice per group). Statistical significance was determined by Student's *t*-test.
(PDF)

**S5 Fig. Efficacy of Isoniazid and Rifampicin treatment.** CMTB was established in mice and after 2 weeks mice were treated for 6 weeks with Isoniazid (0.1 μg/mL) and Rifampicin (0.1 μg/mL) in the drinking water. After treatment, mice were switched to untreated water for at least 1 week to allow complete clearance of the antibiotics from the mice. (The half-lives of Isoniazid and Rifampicin have been measured to be 4 hours and 2.5 hours). Three mice were sacrificed, and undiluted tissue homogenates of the cervical lymph nodes, lungs, and spleens were plated for CFU measurement. No CFUs were detected after the standard 3-week incubation period. The plot shows CFUs measured after 6 weeks of incubation.
(PDF)

**S6 Fig. Correlation between cytokines levels in lungs and spleens of CMTB mice.** CMTB was established as described in the main text and the abundances of selected cytokines and chemokines in the lungs and spleen measured by multiplexed immunoassay at day 42 following inoculation. The plot depicts the levels of analytes that were detected in both tissues. Each point represents the average of 5 mice.
(PDF)

**S7 Fig. Proportions of immune cells in the lungs of control and CMTB mice.** CMTB was established as described in the main text. At 8 weeks, *Left panel*: the absolute number of CD45$^+$ cells in whole-lung homogenates was measured by flow cytometry using counting beads; *Middle panel*: The relative proportions of various immune cell populations were measured by flow cytometry. (See S17 Fig for gating.); *Right panel*: The absolute numbers of CD4$^+$ and CD8$^+$ T cells were measured by flow cytometry using counting beads. (See S14 Fig for gating.) Data are representative of two independent experiments with 4–5 mice per condition.
(PDF)

**S8 Fig. Absolute numbers of AMs in control and CMTB mice following aerosol challenge with Mtb.** Absolute numbers of alveolar macrophages (CD11b$^{int}$CD11c$^+$CD64$^+$Siglec-F$^+$) in control and CMTB mice at 10 days following aerosol infection with 50–100 CFU of Mtb H37Rv. Error bars depict the mean and SD. Representative data from one of two independent experiments with 4–5 mice/group/timepoint. (See S17 Fig for gating strategy.)
(PDF)

**S9 Fig. MHCII expression on pulmonary monocytes from control and CMTB mice following aerosol challenge with Mtb.** MFI of MHCII expression on CD11b$^+$CD64$^+$Siglec-F$^-$ monocytes extracted from control or CMTB mice at 10 days following aerosol infection with 50–100 CFU of Mtb H37Rv. Error bars depict the mean and SD. Representative data from one of two

independent experiments with 4–5 mice/group/timepoint. (See S17 Fig for gating strategy.)
(PDF)

**S10 Fig. Quantification of CD4 T cells in naïve and CMTB mice following aerosol challenge.** Quantification of total and Esat-6-specific CD3$^+$CD4$^+$ cells and in the lung parenchyma (CD45.2-PE$^-$) in control and CMTB mice at 14- and 42-days following aerosol infection with 100 CFU of Mtb H37Rv. A representative cytometry plot of CD3$^+$CD4$^+$ cells is shown. Statistical significance was determined by Student's *t*-Test. Error bars depict the mean and SD. Representative data from one of two independent experiments with 4–5 mice/group/timepoint. (See S14 Fig for gating strategy.)
(PDF)

**S11 Fig. T cells from treated CMTB mice fail to expand upon infection.** (**A**) Following antibiotic treatment to clear CMTB as described in the main text, the fraction of CD3$^+$CD8$^+$ TB10.4$^+$ T cells in the lung vasculature and parenchyma, as determined by i.v. labeling with anti-CD45-PE antibody was measured by flow cytometry 10 days following aerosol challenge with 50–100 CFU of Mtb H37Rv. Statistical significance was determined by Student's *t*-Test. Error bars represent the mean and SEM (**B**) The fraction of CD3$^+$CD8$^+$TB10.4$^+$ T cells in whole lung homogenates of CMTB and control mice prior to aerosol challenge with 50–100 CFU of Mtb H37Rv and at 10 days following challenge was determined by flow cytometry. Statistical significance was determined by paired *t*-test. Error bars depict the mean and SEM. (n = 3–5 mice per group). (See S14 Fig for gating strategy.)
(PDF)

**S12 Fig. CMTB-enhances expression of MHC II on alveolar macrophages in response to aerosol challenge with Mtb and enhanced expression is reduced by antibiotic treatment.** Flow cytometry analysis of alveolar macrophages from control, CMTB, and antibiotic-treated CMTB mice isolated 14 days following aerosol challenge with Mtb H37Rv. CD11b$^{int}$CD11c$^+$ CD64$^+$Siglec-F$^+$ were gated on CD11b and Siglec-F to define AMs (see S17 Fig). MHC II expression was quantified by MFI. Statistical significance was determined by Student's t-Test. Error bars represent the mean and SEM.
(PDF)

**S13 Fig. Distribution of Mtb infection across pulmonary immune cells.** CMTB and control mice were infected with ~2000–4000 CFU of mEmerald-expressing Mtb via aerosol and the cellularity of bronchoalveolar lavage (BAL) fluid extracted 24 hours following infection was analyzed by flow cytometry. Top panels: Flow cytometry plots showing mEmerald+ Mtb-infected AMs gated on CD45$^+$CD11b$^{int}$CD11c$^+$CD64$^+$SiglecF$^+$ cells (see S17 Fig for gating strategy). Bottom panels: Distribution of infected cell types (left) and fraction of AMs infected (right) in BAL fluid at 24 hours following high-dose infection (See Methods). Error bars depict the mean and SD.
(PDF)

**S14 Fig. Bacterial burden in the lung following high-dose aerosol challenge.** Control and CMTB mice were infected with ~2000–4000 CFU of Mtb and bacterial burden in the lung measured by CFU assay 10 days following challenge. Data are representative of 2 independent experiments with 4–5 mice per condition. Significance was assessed by Student's *t*-test. Error bars represent the mean and SD.
(PDF)

**S15 Fig. ATAC-seq on AM from naïve/infected mice at baseline.** Alveolar macrophages from CMTB and control mice (n = 3/condition) were isolated from BAL fluid by FACS and

ATAC-seq was performed following a published protocol [55]. Plot depicts FDR vs. difference in chromatin accessibility between control and CMTB for 45,458 genomic regions. Red dots indicate peaks within gene promotor regions. (See Methods).
(PDF)

**S16 Fig. Lung and peripheral blood T cell gating strategy.** Live (Zombie Violet⁻) single cells were gated on CD3, excluding CD11b⁺, CD11c⁺, Siglec-F⁺, and B220⁺ cells to define T cells and then on CD4 and CD8. For lung samples, localization of T cells to the lung parenchyma or vasculature was determined by i.v. labeling with and anti-CD45.1 antibody.
(PDF)

**S17 Fig. Peripheral blood monocyte gating strategy.** Live (Zombie Violet⁻) single cells were gated on CD45, excluding B220⁺, CD3⁺, NK1.1⁺ and then on CD11b⁺, followed by Ly6G⁻, followed by Ly6C⁻ to define monocytes.
(PDF)

**S18 Fig. Splenic myeloid cell gating strategy.** Live (Zombie Violet⁻) single cells were gated to exclude CD3⁺, B220, and NK1.1⁺. This population was then gated on Ly6Gˡᵒʷ and then on CD11b⁺, CD11cⁱⁿᵗ followed by Ly6c⁻ to define monocytes.
(PDF)

**S19 Fig. Lung myeloid cell gating strategy.** Live (Zombie Violet⁻) single cells were gated on CD11b and Ly6G to define neutrophils (CD11b⁺ Ly6G⁺). Ly6G⁻ cells were further gated on Siglec-F and CD11b to isolate eosinophils (Siglec-F⁺ CD11bʰⁱᵍʰ), alveolar macrophages (Siglec-F⁺ CD11bᵐⁱᵈ), monocytes/interstitial maccrophages and dendritic cells (Siglec-F⁻). In order to robustly isolate alveolar macrophages in inflamed lungs, Siglec-F⁺ CD11bᵐⁱᵈ cells were further gated on CD11c and CD64.
(PDF)

**S20 Fig. Peripheral blood monocyte gating strategy in WT/*Ifngr1*⁻/⁻ mixed bone-marrow chimeras.** Live (Zombie Violet⁻) single cells were gated on CD11b+, excluding CD3, B220, and NK1.1. Monocytes were defined from this population as SSClᵒʷ, Ly6G- cells and their genotypes assigned by CD45.1/2 labeling.
(PDF)

**S21 Fig. Alveolar macrophage gating strategy in WT/*Ifngr1*⁻/⁻ mixed bone-marrow chimeras.** CD3+ and CD19+ cells were excluded from live, single cells to define the myeloid population. Alveolar macrophages were defined from this population as Siglec-F+, CD11c+, CD64 + cells and their genotypes assigned by CD45.1/2 labeling.
(PDF)

**S1 Table. Pathology scores for Mtb-infection of CMTB and control mice.**
(PDF)

**S2 Table. Antibodies used.**
(PDF)

## Acknowledgments

We thank the staff from Seattle Children's Research Institute for animal care and Dr. Piper Treuting and Brian Johnson, University of Washington Immunohistochemistry core, for histological services. We acknowledge the NIH Tetramer Core Facility (Contract HHSN2722 01300006C) for providing of tetramers. We thank Martina Sanlorenzo and Igor Vujic for help with the melanoma model.

## Author Contributions

**Conceptualization:** Johannes Nemeth, Kevin B. Urdahl, Elizabeth S. Gold, Alan Aderem, Alan H. Diercks.

**Data curation:** Dat Mai, Alan H. Diercks.

**Formal analysis:** Johannes Nemeth, Ana N. Jahn, Fergal J. Duffy, Elizabeth S. Gold, Alan H. Diercks.

**Funding acquisition:** Kevin B. Urdahl, Elizabeth S. Gold, Alan Aderem, Alan H. Diercks.

**Investigation:** Johannes Nemeth, Gregory S. Olson, Alissa C. Rothchild, Ana N. Jahn, Dat Mai, Jared L. Delahaye, Courtney R. Plumlee.

**Methodology:** Sanjay Srivatsan.

**Supervision:** Kevin B. Urdahl, Alan Aderem, Alan H. Diercks.

**Validation:** Dat Mai.

**Writing – original draft:** Johannes Nemeth, Gregory S. Olson, Kevin B. Urdahl, Elizabeth S. Gold, Alan H. Diercks.

**Writing – review & editing:** Johannes Nemeth, Gregory S. Olson, Kevin B. Urdahl, Elizabeth S. Gold, Alan H. Diercks.

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
