## [Decision Letter · Decision Letter 0]

8 May 2020

Dear Dr. Diercks,

Thank you very much for submitting your manuscript "Contained Mycobacterium tuberculosis infection induces concomitant and heterologous protection" for consideration at PLOS Pathogens. As with all papers reviewed by the journal, your manuscript was reviewed by members of the editorial board and by several independent reviewers. The reviewers appreciated the attention to an important topic. Based on the reviews, we are likely to accept this manuscript for publication, providing that you modify the manuscript according to the review recommendations.

All three reviewers felt this work will be a valuable contribution, but all provided suggestions for revision.  Some new experiments were suggested to further define mechanism,  However, upon balancing the reviews, the editors believe that all reviewer critiques can be addressed with revisions to text, analyses and conclusions.  

Sincerely,

Christopher M. Sassetti

Associate Editor

PLOS Pathogens

JoAnne Flynn

Section Editor

PLOS Pathogens

Kasturi Haldar

Editor-in-Chief

PLOS Pathogens

orcid.org/0000-0001-5065-158X

Michael Malim

Editor-in-Chief

PLOS Pathogens

orcid.org/0000-0002-7699-2064

Reviewer Comments (if any, and for reference):

Reviewer's Responses to Questions

**Part I - Summary**

Reviewer #1: In this manuscript entitled ‘Contained Mycobacterium tuberculosis infection induces concomitant and heterologous protection’, Nemeth J et al have demonstrated that after subcutaneous (Ear) Mtb infection, the bacteria persisted in the draining lymph node for one year and generated protection in mice re-exposed to Mtb via aerosol route. Interestingly, Mtb infection in the ear also provided heterologous protection against systemic Listeria infection as well as melanoma. The protection in CMTB mice was linked to the low level of circulating IFN-�, activated alveolar macrophages (higher expression of MHCII), and the recruitment of Mtb-specific T cells to the lung parenchyma. The reviewer agrees with the initial assessment of Dr. Ernst that this is an important study and provides more insights to natural immunity against TB. However, the conclusion of the current study needs to be solidified by additional experiments.

Reviewer #2: In this manuscript, Nemeth et al., describe a new model for chronic, low-grade Mtb infection in mice that is established after injection of Mtb bacilli into the ear and is maintained for more than a year. This chronic, low-grade infection induces low-level systemic inflammation that appears to activate alveolar macrophages and that can protect against reinfection with Mtb by aerosol challenge. This effect, which was abrogated in the absence of IFNgR, demonstrating that it is dependent on IFNg signaling, extended beyond mycobacteria and also protected against Listeria and melanoma cells.

The paper reports a novel and interesting model and the first insights into the possible mechanistic underpinnings of concomitant immunity in Mtb, which may be analogous to that observed in humans with controlled, quiescent Mtb infection. The manuscript is well written and the findings are illustrated clearly and are easy to follow. I do not have any major concerns beyond those raised by the previous round of reviews for eLife, which I think were addressed well. The paper may be improved by addressing some minor points.

Reviewer #3: In this manuscript, Nehmes and colleagues describe a chronic ear infection model of Mtb that allows the authors to assess the impact of chronic contained infection on subsequent pulmonary exposure. While not perfectly similar to human “latent” infection, this represents a creative attempt to expand the suite of animal models of tuberculosis.

This manuscript was submitted with reviews from ELife where it seems to have been rejected on the basis of Reviewer 2’s critique that the work was not sufficiently novel or mechanistic. I strongly disagree with this assessment. There is no equivalent small animal model. Moreover, as Dr. Ernst says as Reviewer 1, novel observations from well executed studies are a critical foundation for subsequent mechanistic work –and probably on average hold up better over time.

I do have series of more granular concerns of which the final one is the most serious:

-Low dose challenge model: I have concerns about the analysis. Protection is manifested as reduction in “take” –the number of bacteria that infect where the answer can be 0,1,2 or 3 and subsequent growth/control of those bacteria, which is exponential-ish. First, the methods say that the authors excluded counts from mice that had no CFU. I am not sure that it is legal to exclude the zeros and it obscures biology where showing in up in 1 vs 2 lobes might be an effect on take or spread.

-Heterologous immunity: I disagree with the interpretation of the Leishmania and melanoma studies. I think that these studies indicate that CMTB provides heterologous and perhaps innate protection against these two challenges but it does not necessarily follow that the protection against Mtb is through the same mechanism and likely to be innate.

-Unappreciated T cell responses: The authors seem to favor innate mechanisms of control but their data strongly suggest the contribution of a primed T cell response and less convincing makes a case for myeloid memory. Indeed, I am not sure why the authors do not think the differences that they see in AM transcriptional responses post challenge (the IFN signature) do not reflect the consequences of a primed T cell response. Similarly, I am not sure how pure the sorted AM populations were but a little T cell contamination with some associated Ifng production could explain the apparent differences in AM ability to restrict Mtb (4H)

**Part II – Major Issues: Key Experiments Required for Acceptance**

Reviewer #1: 1) The authors have provided tantalizing data of how innate immunity in particular AM play an important role in protection against pulmonary Mtb in CMTB mice. However, the cellular mechanisms of cross-protection of CMTB in listeria or melanoma models have not been addressed. If the authors assume that potentially the bone marrow derived monocytes/macrophages in CMTB were providing better protection in these model systems, can they simply assess the inflammatory capacity of BMDM (CMTB vs Control) in ex-vivo model of PAMPs stimulation (Similar to Figure 3B). Moreover, the authors can also treat these BMDM in vitro with a low dose of IFN-g to recapitulate the potential impact of low circulating IFN-g on these cells and their protection capacity against subsequent Mtb or listeria infection (in vitro CFU).

2) While the investigators suggest that IFN-g signaling was required for the activation marker (MHC-II) of monocytes and AM in mixed BM chimeras, the impact of this reduction on protection against Mtb was not demonstrated. What is the level of CFU in the lungs/spleens of these chimeric mice after Mtb infection? It is also intriguing that the pathway analysis reveals type I IFN (IFN-a) was highly induced in CMTB (Figure 4E). Can the authors perform the same mixed BM chimeras using IFNAR KO mice? The comparison of type I vs type II would be of interest in this model system. Hence several studies suggest that type I IFN is detrimental in immunity to TB, while it has been shown that IFN-a play a key role in regulating macrophages-mediating immunopathology in influenza viral infection (PMID:31110361).

3) As all the human BCG vaccination is intradermally, the study will benefit from comparison of CMTB to intradermal (Ear) BCG vaccination (CBCG!) to see how potentially a virulent strain vs an avirulent strain may affect trained immunity against subsequent pulmonary Mtb infection.

Reviewer #2: (No Response)

Reviewer #3: above

**Part III – Minor Issues: Editorial and Data Presentation Modifications**

Reviewer #1: 1) The vast majority of LTBI have been initially infected with Mtb via pulmonary route and thus their AM are not “naïve” and have been exposed to Mtb. In fact, this comparison is significantly weakened their central hypothesis as the authors indicated that the presence of Mtb in the skin draining lymph nodes and not lung initiates a strong immunity to pulmonary Mtb infection. Thus, the reviewer highly encourages the authors of not comparing this model system (CMTB) to latent TB infection in humans.

2) Can the authors provide some information of inflammatory responses at the initial site of infection (ear)?

3) Alveolar macrophages from CMTB mice showed higher expression of MHC II. Did the author study the expression of MHCII on other CD11b+ cells in CMTB mice both pre and post aerosol Mtb infection? It’s interesting to assess the expression of MHCII or other activation marker on CD11b+ cells.

4) CMTB mice showed more recruitment of monocytes derived macrophages in the lungs 14 days post Mtb aerosol infection. Is there any change in the frequency or total AM in the lungs pre and post aerosol Mtb infection?

5) What is the source of IFN-� in the lung of CMTB mice?

Reviewer #2: 1. The results of statistical analyses are reported as asterisks that represent categories of p-values. It would be more appropriate if the reader can interpret these results based on the actual p-value. As such, wherever possible the authors should show actual p-values (even when these are above 0.05, or whatever threshold was used) instead of asterisks.

2. The authors claim that their results provide insights into new vaccine development approaches, but do not discuss how their results can be used to advance this field. The authors should augment their discussion by adding some text about how the findings can be used to improve vaccination approaches.

Reviewer #3: above

PLOS authors have the option to publish the peer review history of their article (what does this mean?). If published, this will include your full peer review and any attached files.

Reviewer #1: No

Reviewer #2: No

Reviewer #3: Yes: Sarah Fortune
---

## [Editor Report · Decision Letter 1]

26 May 2020

Dear Dr. Diercks,

We are pleased to inform you that your manuscript 'Contained Mycobacterium tuberculosis infection induces concomitant and heterologous protection' has been provisionally accepted for publication in PLOS Pathogens.

Best regards,

Christopher M. Sassetti

Associate Editor

PLOS Pathogens

JoAnne Flynn

Section Editor

PLOS Pathogens

Kasturi Haldar

Editor-in-Chief

PLOS Pathogens

orcid.org/0000-0001-5065-158X

Michael Malim

Editor-in-Chief

PLOS Pathogens

orcid.org/0000-0002-7699-2064
---

## [Editor Report · Acceptance letter]

15 Jun 2020

Dear Dr. Diercks,

We are delighted to inform you that your manuscript, "Contained *Mycobacterium tuberculosis* infection induces concomitant and heterologous protection," has been formally accepted for publication in PLOS Pathogens.

Best regards,

Kasturi Haldar

Editor-in-Chief

PLOS Pathogens

orcid.org/0000-0001-5065-158X

Michael Malim

Editor-in-Chief

PLOS Pathogens

orcid.org/0000-0002-7699-2064